# Update on the Pathogenesis of the Hirschsprung-Associated Enterocolitis

**DOI:** 10.3390/ijms24054602

**Published:** 2023-02-27

**Authors:** Shuai Li, Yichun Zhang, Kang Li, Yuan Liu, Shuiqing Chi, Yong Wang, Shaotao Tang

**Affiliations:** Department of Pediatric Surgery, Union Hospital, Tongji Medical College, Huazhong University of Science and Technology, Wuhan 430022, China

**Keywords:** Hirschsprung disease (HSCR), enterocolitis, Hirschsprung-associated enterocolitis (HAEC), pathogenesis, review

## Abstract

Despite the significant progress that has been made in terms of understanding the pathophysiology and risk factors of Hirschsprung-associated enterocolitis (HAEC), the morbidity rate has remained unsatisfactorily stable, and clinical management of the condition continues to be challenging. Therefore, in the present literature review, we summarized the up-to-date advances that have been made regarding basic research on the pathogenesis of HAEC. Original articles published between August 2013 and October 2022 were searched in a number of databases, including PubMed, Web of Science, and Scopus. The keywords “Hirschsprung enterocolitis”, “Hirschsprung’s enterocolitis”, “Hirschsprung’s-associated enterocolitis”, and “Hirschsprung-associated enterocolitis” were selected and reviewed. A total of 50 eligible articles were obtained. The latest findings of these research articles were grouped into gene, microbiome, barrier function, enteric nervous system, and immune state categories. The present review concludes that HAEC is shown to be a multifactorial clinical syndrome. Only deep insights into this syndrome, with an accrual of knowledge in terms of understanding its pathogenesis, will elicit the necessary changes that are required for managing this disease.

## 1. Introduction

Hirschsprung-associated enterocolitis (HAEC) is the leading cause of serious morbidity and mortality in patients with Hirschsprung disease (HSCR) [1]. Clinically, among the various risk factors, delayed diagnosis, the type of operation employed, female sex, having a younger age at presentation, long-segment disease, family history, associated anomalies, and anastomotic leaks or strictures are the most commonly reported in the literature [2]. One of the leading causes or risk factors for HAEC has been considered to be partial mechanical obstruction. The major underlying feature of this may be anastomotic stricture, bowel-disordered motility, or functional obstruction (paralysis). There is some evidence to suggest that recurrent HAEC may be released following internal sphincter myotomy [3]. However, HAEC also occurs in patients with enterostomy, and without any evidence of obstruction. Therefore, other factors must be involved.

Given the numerous advances that have been made in terms of performing meticulous operations procedures, standardized diagnostic systems [4], and postoperative management [5], the reported incidence of HAEC remains stable, ranging from 6 to 60% prior to pull-through surgery, and from 25 to 37% following surgery [4]. Several studies have found that the proximal dilated segment of the colon is mostly affected and more susceptible to HAEC [6,7,8], which seems to explain part of the reason why HAEC still occurs postoperatively. In addition, some studies have found that older age at radical surgery is a risk factor for the development of postoperative HAEC [9], and older age is also a risk factor for the development of preoperative HAEC [10]. However, the data concerning HAEC etiology remain limited.

On the other hand, since the comprehensive review by Demehri [11], basic research on HAEC has made marked progress in terms of increasing the knowledge base, especially with respect to studies on genes, the microbiome, immunity, and other aspects [12,13]. In the current review, we summarize the advances (Figure 1) that have been made in understanding HAEC pathogenesis over the course of the last decade, with the intention of enabling researchers to gain a better grasp of the current status of essential knowledge on HAEC, which should be of use in terms of planning future research directions.

## 2. Genes

Previous studies have revealed that genetic background influences HAEC, mainly in the form of changes in the incidence and severity of HAEC in patients with HSCR with a combination of several clinical syndromes. Kwendakwema [14] conducted a retrospective cohort study of 207 patients with HSCR, 26 (13%) of whom were trisomy 21 (T21) patients, and found that the incidence of HAEC in children with HSCR and T21 (38%) was not significantly different from that in children with HSCR alone (41%). By contrast, Halleran’s study found that, compared with patients with HAEC alone, patients with a combination of HAEC and T21 experienced more severe symptoms, including longer duration of symptoms, hypotension, greater likelihood of tachycardia and longer times on antibiotics, and were also more likely to require intensive care unit admission [15]. A higher incidence of HAEC in children with Mowat-Wilson syndrome with HSCR has also been reported [16].

The advent of whole-exome sequencing has also facilitated this type of research. Bachetti [17] identified p.H187Q in the oncostatin-M (OSM) receptor (OSMR) gene as a susceptibility variant of HAEC. It may exert a key role in HAEC pathogenesis by regulating/activating the OSM-OSMR *axis*. Via large-sample sequencing, the single nucleotide polymorphisms (SNPs) rs8104023 [18] and rs2191026 [19] were found to be significantly associated with postoperative HAEC. In another study, DNA was extracted from the colon tissue samples of 30 patients with HAEC, and the mRNA expression of integrin beta-2 (ITGB2; also known as CD18) was found to be negatively correlated with the incidence and severity of HAEC [20].

Taken together, the above studies have suggested that the pathogenesis of HAEC is closely associated with the underlying genes and the genetic background (Figure 2), although the specific mechanism(s) involved still require further study.

## 3. Intestinal Microbiome

The human intestinal microbiome is a complex ecosystem in which the phyla Firmicutes and *Bacteroidetes* are dominant, followed by proteobacteria [21], and the normal intestinal flora is in a dynamic balance. Maintaining the relative balance of the intestinal microflora is closely associated with the stability of the internal environment of the intestine and its normal function.

### 3.1. Animal Experimental Basis for the Relevance of Gut Microbes to HAEC

Pierre [22] found that mice with neural crest conditional deletion of endothelin receptor B (EdnrB) exhibited impaired mucosal barrier function and developed ecological dysregulation prior to the onset of HAEC, as evidenced by decreased levels of luminal secretory phospholipase A2 (sPLA2) and increased intestinal invasion by *Escherichia coli* prior to HAEC and death. Cheng [23] found that the proportion of fecal bacteria in HAEC model mice increased in the case of *Akkermansia* and decreased in the case of *Bacteroidetes*, and that the genera with reduced abundance in HAEC were *Dysgonomonas* and *Clostridium cluster* XIVa, suggesting that *Akkermansia* may contribute to the development of HAEC, whereas *Bacteroidetes*, *Dysgonomonas,* and *Clostridium cluster* XIVa may exert a protective effect. It has been shown that the abundance of *Veillonella parvula* (VP) in intestinal microorganisms is higher in patients with HAEC compared with patients with HSCR [24]. Zhan [25] found that an excessive density of VP exerted proinflammatory effects by increasing the concentration of inflammatory cytokines and impairing intestinal motility in the colon, and VP-derived lipopolysaccharide (LPS) was used to establish a mouse model of inflammation wherein LPS both elicited a markedly enhanced paracellular permeability of mouse colonic epithelial cells and activated macrophages via the Toll-like receptor 4 (TLR4) pathway. Tumor necrosis factor-α (TNF-α) from polarized macrophages was found to impair the pacemaker function of interstitial cells of Cajal (ICCs), which subsequently inhibited intestinal motility, and intestinal dysmotility exacerbated intestinal dysbiosis, in turn promoting the development of HAEC. In another study, Mitroudi [26] cultured *E. coli*, *Enterococcus* spp., *Bacillus*, *Proteus mirabilis*, and *Clostridium* spp. in the mesenteric lymph nodes, spleen, liver, kidneys, and lungs of HACE-modeled rats, and found that *E. coli* had the highest ectopic rate. All the rats sacrificed 25 days after modeling exhibited live *E. coli* in extraintestinal sites, whereas the control and sham group rats did not. Although the rodent models described above differed from human physiology, Arnaud [27] found that proinflammatory bacteria (*Bilophila* and *Fusobacterium*) were more abundant in the aganglionic rectosigmoid lumen. However, it is not clear whether this situation arose consequentially after the onset of the disease, or whether it was a contributing factor toward it.

### 3.2. Clinical Studies on the Relevance of Intestinal Microorganisms to HAEC

Searching for risk pathogens has also been performed in a clinical setting. Parker [28] collected stool samples of patients with HAEC at different time points from the onset of symptoms to remission, and analyzed changes in the bacterial community therein; a higher abundance of *Blautia* was observed in the remission samples. *Clostridium difficile* is also considered to be a risk pathogen for HAEC [29].

It has been hypothesized that higher biodiversity could also have a role in maintaining intestinal homeostasis, and that its disruption could promote the development of HAEC [30,31]. By isolating fecal DNA, Frykman [30] observed reduced gut microbial diversity in patients with HAEC. Patients with total colonic aganglionosis, a condition that has the highest risk of HAEC, had more proteobacteria and a lower diversity of gut microbiota compared with patients with rectosigmoid aganglionosis [31]. Several studies have found significant microbiota differences when comparing between patients with HSCR and patients with HAEC, and increased populations of intestinal proteobacteria in patients with HAEC, through the use of Illumina-MiSeq sequencing [24,30,32]. Similarly, a comparison of the flora after the onset of symptoms does not confirm whether the abnormal flora is a cause or a consequence of HAEC, and therefore prospective studies are required in this regard.

In a different study, Yan [32] collected samples of the intestinal contents from four patients from different sites along the intestine during surgery and isolated their DNA, also using Illumina-MiSeq sequencing. In their study, it was found that patients with HAEC exhibited greater bacterial diversity compared with patients with HSCR, in contrast with the previous studies of Frykman and Prato [30,31]. The majority of the clinical studies published to date have collected stool samples from patients for sequencing and used the results to represent the intestinal flora, whereas a minority of studies have collected surgical intestinal specimens for sequencing during surgery. Which of these methods is superior in terms of reflecting the real situation of intestinal mucosal microorganisms, in addition to the biological and clinical association with HAEC, needs to be addressed in future studies.

An important physiological role of intestinal bacteria is the production of short-chain fatty acids (SCFAs), and SCFAs serve an important role in maintaining the integrity of the colonic mucosa [33]. Demehri found that stool SCFA levels were reduced >4-fold in children with a history of HAEC, and that the SCFA composition was altered, perhaps suggesting a complex interaction between colonic metabolism and the microbiota changes [34]. In another series of studies, Plekhova [35] compared fecal metabolites in patients with or without a history of HAEC, and found that patients with HAEC exhibited increased tyrosine catabolism and elevated levels of trans-4-hydroxy-l-proline (Hyp) and 4-methyl-3-penten-2-one, as well as decreased levels of ethyl pentyl ketone. Tyrosine-based compounds act as signaling molecules between the microbiome and the host [36]. Increased concentrations of Hyp may signify a decrease in the Hyp-utilizing microbiota. Furthermore, 4-methyl-3-penten-2-one and ethyl pentyl ketone can originate from the microbiome and the metabolites of the host. This implies that metabolic abnormalities may be caused by dysregulated gut microbes, which in turn further promotes colonic ecological dysregulation and, finally, HAEC development.

In addition, a prospective cohort study conducted by Tang [37] found that patients with HSCR who were exclusively breastfed had lower numbers of Gram-negative bacteria, especially Enterobacteriaceae, and low concentrations of LPS, and were, therefore, less prone to HAEC; these findings led the authors to speculate that exclusive breastfeeding to regulate the gut microbiota may reduce endotoxin biosynthesis and release, thereby preventing patients with HSCR from progressing to HAEC.

Considered together, the above studies suggest that HAEC is closely associated with the intestinal microflora (Figure 2), but whether abnormal microflora is a cause of the disease progression, is a consequence of it, or merely has a participatory role needs to be addressed in further prospective studies.

## 4. Intestinal Mucosal Barrier

The mucosal barrier serves as the first line of defense, protecting the healthy intestinal surface from adhesion and invasion by tubular microorganisms. The components of the intestinal barrier include the lumen, microenvironment or mucus-containing layer, epithelium, and lamina propria. Numerous studies have shown that structural defects and dysfunction of the intestinal mucosal barrier are responsible for the pathogenesis of HAEC.

### 4.1. Altered Mucosal Barrier Properties

Several studies have analyzed the passive diffusion of particles to assess the mucus layer barrier function in the mouse colon, and these studies identified that the diffusion rate in the colonic mucosa of EdnrB^−/−^ mice was significantly reduced compared with the wild-type mice [38,39]. Yildiz [39] found that the efficiency of active microbial transport and passive diffusion of granular material in the colonic mucosa of EdnrB^−/−^ mice were also significantly reduced. Dariel [40] collected surgically resected intestinal samples from short-segmented HSCR neonates, and found that the intestinal epithelial barrier (IEB) permeability was significantly increased in the ganglia of patients with HSCR who developed HAEC postoperatively, which suggested that abnormal IEB is closely associated with HAEC.

### 4.2. Goblet Cells (GCs)

GCs are specialized secretory cells that are located throughout the mucosal epithelium of the intestinal tract, and form the main components of the mucus layer through the secretion of gel acting against pathogen infection [41].

Mucin 2 (MUC2) is the major mucin expressed in humans [42]. Trefoil factor 3 (TFF3) acts synergistically with mucins to enhance the protective barrier properties of the mucus layer [43]. SAM-pointed domain-containing ETS-like factor (SPDEF) drives the terminal differentiation and maturation of secretory progenitors into GCs [44]. Kruppel-like factor 4 (KLF4) is a GC-specific differentiation factor in the colon, which serves to both regulate GC differentiation and activate mucin synthesis [45]. Significant downregulation of the expression levels of TFF3, SPDEF, and KLF4, and a significant reduction in the GC population, were observed in the aganglionic and ganglionic colon of patients with HSCR [46], factors which may increase the susceptibility of patients to HAEC.

However, Thiagarajah [38] discovered that, prior to the development of HAEC, patients with HSCR had increased numbers of colonic GCs, which were similar to the size of the GCs of non-HSCR controls, but reduced levels of neutral and acidic mucins derived from the GCs. EdnrB^−/−^ mice exhibited both an increased size and number of colonic GCs in distal aganglionic segments, whereas, in proximal ganglionic segments, the size and number of the GCs were decreased both in patients with HSCR and EdnrB^−/−^ mice. The level of the membrane-bound mucin Muc4 was reduced, suggesting an altered maturation process for the HSCR GCs. The increased number of GCs identified in this study contradicted the previous results of Nakamura [46], however, and therefore needs to be confirmed in further studies.

Porokuokka [47] identified a 70–80% reduction in the expression level of the mouse glial cell line-derived neurotrophic factor (GDNF) co-receptor, GDNF family receptor alpha-1 (GFRα1), in both HSCR and HAEC. In addition, HAEC was observed in GFRα1 hypomorphic mice that were experiencing GC dysplasia, abnormal mucin production, and storage. However, progression to the conditions of epithelial damage, microbial adhesion, and tissue invasion was only observed at advanced stages in the mice, suggesting that bacterial adhesion was not the initiating factor of HAEC, and that GC dysfunction in the colon had preceded these other pathological changes.

In conclusion, these studies have shown that dysfunction of GCs and reduced secretion of mucin lead to damage of the intestinal mucosal barrier, thereby leading to the development of HAEC. Restoring both normal GC function and the quantity of colon epithelium may be potential targets for preventive therapy of HAEC, and these possibilities should be further investigated from this perspective in the future.

### 4.3. Others

Tight-junction proteins fulfill a key role in regulating epithelial barrier function, and are composed of structural proteins, including claudin protein, occludin protein, junctional adhesion molecules (JAMs), zonula occludens-1 (ZO-1), and various types of ligand-protein molecules. Arnaud [27] observed that the expression of ZO-1 was significantly lower in piglets with hypoganglionic sigmoid, whereas the expression levels of claudin-3 and E-cadherin were increased. By contrast, Dariel [40] identified no significant differences in the expression levels of several tight-junction proteins, including ZO-1, occludin, junctional adhesion molecules A (JAMA), cingulin, and claudin-1, comparing among patients with HAEC, patients with HSCR, and patients with HSCR combined with obstructive symptoms/diarrhea.

ATP-sensitive K^+^ (K_(ATP)_) channels have been described as passive transducers for ions. Previous studies have identified decreased expression levels of K_(ATP)_ channels in HSCR specimens, and co-localization of Kir6.1 and SUR2 (subunits of K_(ATP)_ channels) with the tight-junction protein claudin-1, suggesting that alterations in K_(ATP)_ expression may both affect intestinal epithelial integrity and be associated with the pathogenesis of HAEC [48]. TREK-1 (also termed potassium channel subfamily K member 2 (KCNK2)) is a mechanosensitive K2P channel, and Tomuschat [49] found reduced expression levels of TREK-1 in both the ganglionic and aganglionic regions in HSCR; moreover, a deficiency of TREK-1 was reported to induce barrier dysfunction in human colonic epithelial cells [50].

The TLR4/phosphorylated (p)-p38/nuclear factor-kappaB (NF-κB) signaling pathway in the intestine, and F-actin expression in the intestinal epithelial cytoskeleton, are important for maintaining both intestinal mucosal integrity and intestinal barrier function [51,52]. Zheng [53] infected EdnrB^−/−^ mice with *E. coli* JM83 cells via oral gavage in order to establish an HAEC model. The EdnrB^−/−^ mice were found to have significantly elevated expression levels of TLR4, NF-κB, and p-p38, substantially reduced cytoskeletal F-actin protein density, and severely disrupted tight junction structures. Protease-activated receptors (PARs)-1 and -2 are known to be associated with intestinal permeability, regulation of intestinal motility, and inflammatory reactions. Increased expression of PAR-1 and PAR-2 and an excessive local release of PAR-activating proteases were observed in Tomuschat’s study in the colon of patients with HSCR [54].

In conclusion, intestinal mucosal barrier dysfunction and tissue structural disruption have been shown to be closely associated with the development of HAEC (Figure 2). However, the fundamental research on GCs and tight-junction proteins remains limited.

## 5. Enteric Nervous System (ENS)

The abnormal ENS of HSCR leads to impaired intestinal motility, resulting in functional obstruction with subsequent stasis and overgrowth of pathogenic intestinal bacteria, destruction of the mucosal layer, invasion of the intestinal wall, dysfunction of the intestinal mucosal barrier, impaired immune response and consequent HAEC. This abnormal neurological function is considered to be an important cause of preoperative HAEC episodes.

### 5.1. Cholinergic Neurons

Acetylcholine is an important neurotransmitter that promotes intestinal motility. Keck [55] found that a low degree of colonic mucosal acetylcholine-positive innervation led to an enhanced inflammatory immune cell state, disrupted microbial metabolism, and a higher incidence of postoperative HAEC. A similar phenomenon was observed in another animal experiment performed by Porokuokka [47]. In addition, Feng [8] found a reduced acetylcholine content and a higher level of inflammation in the dilated segment compared with the HSCR segment in EdnrB^−/−^ mice. Taken together, these studies suggest that a correlation exists between acetylcholine and HAEC.

### 5.2. Nitrergic Neurons

Nitric oxide (NO) is an important second messenger and inflammatory mediator that exerts a key role in intestinal barrier failure in numerous types of intestinal inflammatory diseases [56]. NO synthase (NOS) exists in multiple isoforms, including endothelial NOS (eNOS), neuronal NOS (nNOS), and inducible NOS (iNOS). Most of the beneficial effects of NO are produced by the constitutive synthases, eNOS and nNOS, which maintain stable intestinal barrier function and mediate intestinal homeostasis [57]. Several studies have demonstrated a link between NOS and HAEC.

Caveolin-1 (Cav-1) regulates the functions of different NOS isoforms, and Nakamura [58] previously found a lower expression level of Cav-1 in the colon of patients with HSCR, which led them to speculate that reduced Cav-1 expression may lead to excessive activation of iNOS, consequently leading to epithelial damage and thus increasing the susceptibility of HSCR to HAEC. However, by contrast, the results of the study by Dariel [40] showed that Cav-1 expression appeared to be higher in patients with HAEC. It should be acknowledged that a limitation of both these studies was the small number of patients involved, and further studies are required.

NOS-interacting protein (NOSIP) is a modulator of NO production that is able to inhibit the production of NO. Tomuschat [59] found that an increased expression level of NOSIP in patients with HSCR may promote HAEC development by inhibiting the local production of eNOS and nNOS. Additionally, associated with Tomuschat’s study was the novel finding of an increased proportion of nNOS neurons both in patients with HSCR and in HSCR model mice [60].

In Dariel’s study [40], through comparing surgically resected bowel samples from HSCR neonates and the proximal end of the stoma of patients with anorectal malformation (ARM), the number of nNOS enteric neurons in the nerve segments of patients in the HSCR combined with obstructive symptoms group was found to be significantly reduced, whereas the proportion of nNOS to total enteric neurons was higher in patients in the HSCR combined with HAEC or diarrhea groups, similar to the results obtained by Cheng [60]. Taken together, these studies showed that an increased proportion of nNOS to total enteric neurons may lead to postoperative bowel dysfunction in HSCR, also including cases of HAEC; although, Cheng [60] did not make further comparisons based on different classifications of postoperative complications, and so the ability to compare the results in terms of the correlation between nNOS and HAEC is low, and further studies are therefore required to clarify this correlation.

### 5.3. TLR

Research suggests that TLR4 expression can regulate ENS maturation and development [61]. Dariel [40] conducted a prospective multicenter cohort study and found that, compared with the ARM group, the HSCR combined with the HAEC group exhibited significantly lower expression levels of TNF-α, TLR2, and TLR4, increased paracellular and transcellular permeability, and a strong correlation was observed between TLR2/4^−/−^ expression and the number of nNOS myenteric neurons. Furthermore, TLR2^−/−^ and TLR4^−/−^ mice were found to have overall reduced numbers of neurons and nNOS-immunoreactive (IR) neurons [61,62], which enabled Dariel to speculate that a reduced level of TLR2/4 expression may lead to an altered ENS phenotype, and therefore an increased susceptibility to postoperative bowel dysfunction.

However, by contrast, Zheng [53] found significantly higher levels of TLR4, NF-κB, p-p38, TNF-α, transforming growth factor (TGF)-β, and interleukin (IL)-10 expression in EdnrB^−/−^ mice compared with wild-type mice. TLR4 knockdown led to a reduction in the severity of small intestinal colitis, and the expression levels of IL-10, TNF-α, and TGF-β were all attenuated. These results appeared to differ from those of Dariel [40], although the subjects and controls of the two studies were different, and this may have contributed to the different results; additional follow-up experiments are therefore required in the future to further confirm this.

### 5.4. ICCs

ICCs, located between the plexuses, are considered to act as the pacemakers of gastrointestinal peristalsis, regulating the activity of the intestinal muscles [63]. Jankovic [64] found lower numbers of ICCs in both the transitional and normoganglionic zones of children with HSCR compared with normal controls, and these numbers were lower still in patients with postoperative constipation and enterocolitis. Another study found that, in the dilated colon of both patients with HAEC and EdnrB^−/−^ mice, ICCs lost the c-Kit phenotype, leading to impaired pacemaker function and impaired intestinal motility [6]- findings that were consistent with those of Zhan [25]. Taken together, these studies suggest that a decreased population of ICCs and loss of phenotypic expression lead to both suppressed and impaired intestinal motility and an exacerbation of intestinal flora dysbiosis, which, in turn, promotes the development of HAEC.

Hydrogen sulfide, synthesized from L-cysteine by two key enzymes, cystathionine-β-synthase (CBS) and cystathionine-γ-lyase (CSE), has been reported to have a key role both in regulating gastrointestinal motility and in promoting the resolution of inflammation [65]. Tomuschat [66] has shown that the expression levels of CBS and CSE in smooth muscle, ICCs, platelet-derived growth factor-α receptor-positive cells, enteric neurons, and colonic epithelium were markedly decreased in HSCR specimens, indicating that mucosal integrity and colonic contractility may have been affected, thereby rendering patients with HSCR more susceptible to developing HAEC.

In conclusion, HAEC may be associated with reduced cholinergic innervation in the intestinal mucosa, a reduced density of NOS enteric neurons, abnormal receptor expression leading to abnormal neural development, reduced numbers of ICCs, and their phenotypic loss (Figure 3).

## 6. Immune System

Intestinal-associated lymphoid tissue is the largest immune organ in the body. Macrophages that have colonized the intestine are the first line of defense of the intestinal immune response, and serve to protect the intestine from pathogenic microorganisms. B lymphocytes mature and differentiate into plasma cells that secrete secretory immunoglobulin A (sIgA) into the intestinal lumen, which is the main immunoglobulin in the intestine and is actively transported to the mucosal surface as a dimer through the polymeric immunoglobulin receptor (pIgR), thereby maintaining the balance and normal function of the intestinal microenvironment [67].

### 6.1. Immune Organs

Gosain [68] found that, compared with EdnrB^NCC+/−^ mice, EdnrB^NCC−/−^ mice (i.e., mice with a conditional neural crest-specific deletion of EdnrB) had smaller spleens and a lower proportion of spleen weight to total weight. Furthermore, Frykman [69] found that HAEC mice showed thymus degeneration and splenic lymphoid reduction. However, in this case, the maturation and functional markers of the immune organs were not investigated, and the underlying mechanism(s) that would account for how the immune organs are involved in HAEC remain unclear.

### 6.2. Immune Cells

As is already known, immune cells are heavily involved in the pathogenesis of HAEC. Macrophage phenotypes, including the classically activated (M1) and alternatively activated (M2) types, have been well studied. During the early stages of inflammation, macrophages are predominantly of the M1 type, promoting the development of inflammation, whereas M2 macrophages promote tissue stabilization and maturation [70]. In the proximal dilated colon of both patients with HAEC and EdnrB^−/−^ mice, significant infiltration of proinflammatory M1 macrophages was observed, whereas the population of M2-type macrophages was higher in the distal colon compared with the proximal segment and the function of the ICCs was impaired [6], suggesting that M1 macrophages may contribute to the development of HAEC by causing loss of phenotype of the ICCs and consequently pacemaker function, leading to intestinal dysmotility and further HAEC. Zhan [25] found that VP released LPS in vivo to activate M1 macrophages through the TLR4 pathway, and M1 macrophages produced TNF-α, which in turn promoted the development of HAEC. This study also found that 4-octyl itaconate (OI) could reduce the production of proinflammatory factors and promote the recovery of the ICC phenotype via inhibiting macrophage activation, a discovery that may provide a potential HAEC therapeutic direction in the future. It has been shown that HAEC mainly occurs in the dilated segment of the HSCR mouse model, where the content of acetylcholine in the stenosis segment is increased, and inflammation occurs at a relatively low level. Acetylcholine acts on the α7 nicotinic acetylcholine receptor (α7nAChR) on the surface of macrophages to inhibit macrophage activation, thereby activating the JAK2-STAT3 anti-inflammatory pathway and inhibiting the NF-κB inflammatory pathway [7,8]. The above studies suggest that multiple pathways are involved in promoting the progression of HAEC through the activation of M1 macrophages, and these may exert a significant role in the development of HAEC.

Lymphocytes are essential components of the immune system, are widely distributed in the body, and recognize antigens to generate specific immune responses. Keck [55] observed a decrease in Treg cells and an increase in Th17 cells in colon tissue dominated by low cholinergic fibers with a higher incidence of HAEC. Gosain [68] identified both a defect in B lymphocyte maturation and a reduction in their numbers in EdnrB^NCC−/−^ mice. The deficiency in antibody production of the B lymphocytes was confirmed in the HAEC mouse model [64]. Frykman [69] found that EdnrB^−/−^ mice and Edn3 ligand-knockout mice (Edn3^−/−^ mice) had significantly reduced numbers of T cells; common lymphocyte progenitor populations were also significantly reduced, and B lymphocyte production was suppressed, with the degree of suppression being strongly correlated with the severity of HAEC. Taken together, these findings suggest that reduced numbers, or functional defects, of lymphocytes may lead to an attenuated specific immune response in the gut and, consequently, increased susceptibility to HAEC.

### 6.3. Antibodies

Immunoglobulin A (IgA) is the main antibody that protects the mucosal surface of the body by binding specifically to the surface structures of pathogens, thereby blocking their adhesion to the mucosa, and preventing the occurrence of infection [71]. Gosain [68] found reduced levels of IgA secretion in the small intestine of EdnrB^NCC−/−^ mice compared with EdnrB^NCC+/−^ mice, whereas the level of nasal and bronchial IgA secretion was unchanged, suggesting an intestinal-specific defect in either IgA production or secretion. Medrano [72] found that, compared with wild-type mice, EdnrB^NCC−/−^ mice exhibited reduced production levels of IgA, IgG, and IgM, and a 50% decrease in small-intestinal pIgR following the in vitro stimulation of splenic B lymphocytes.

In addition to animal experiments, an association between antibodies and HAEC has also been observed in clinical studies. Frykman [73] used an enzyme-linked immunosorbent assay to detect inflammatory bowel disease-associated antibodies, namely anti-*Saccharomyces cerevisiae* antibody (ASCA), anti-*E. coli* outer membrane porin C (OMPC), anti-flagellin (CBir1), and antineutrophil cytoplasmic antibodies (ANCA), in the plasma of children with HSCR, and found that the serum OMPC antibody and ASCA IgA levels were elevated in patients with HAEC, and that elevated OMPC antibody levels were associated with the development of HAEC, suggesting that HAEC and Crohn’s disease share a common intestinal microbial–host immune response, and that these antibodies may be potential biomarkers for the diagnosis of HAEC.

### 6.4. Cytokines

ILs are soluble proteins secreted by cells of the immune system that fulfill important roles in immune regulation. IL-23, a member of the IL-12 family, is an important survival factor for Th17 cells, promoting the secretion of IL-17 by Th17 cells [74]. IL-17 is a proinflammatory cytokine, and the adaptor protein Act1 interacts with the IL-17 receptor (IL-17R) [75]. IL-36γ is a member of the IL-1 superfamily, which is involved in host defense, leading to a proinflammatory response and the development of inflammatory diseases [76]. IL-36 receptor (IL1RL2) is an important mediator molecule in the inflammatory response, and is associated with mucosal repair mechanisms within the colonic epithelium in rodent models of experimental colitis [77]. It was found that the expression levels of Act1, IL-17R, and IL-36γ were significantly increased, whereas that of IL1RL2 was significantly decreased, in HSCR specimens, and that the Th17 cell-associated cytokines IL-17 and IL-23 were upregulated in HAEC [78,79,80]. Keck [55] observed an increased expression of IL-23 in colonic tissue innervated by hypocholinergic fibers, where the incidence of HAEC is higher.

TNF-α is an important proinflammatory factor and immunomodulatory factor that is mainly produced by M1 macrophages and, in turn, promotes the development of HAEC [25]. Another study performed by Chen showed that TNF-α expression was significantly increased in more inflamed dilated segments compared with stenotic segments [8]. Meng [81] found that, compared with wild-type mice, not only was intestinal inflammation more severe, but the plasma levels of the proinflammatory factors TNF-α and interferon (IFN)-γ were significantly higher, whereas those of the anti-inflammatory factors TGF-β and IL-10 were significantly lower, in an HAEC mouse model. Chen [6] found that the level of TNF-α was significantly increased, and ICCs lost their c-Kit phenotype, in the proximal dilated colon of both patients with HAEC and EdnrB^−/−^ mice. TNF-α-mediated p65 phosphorylation was shown to induce miR-221 overexpression, leading to inhibition of c Kit expression and pacemaker currents, suggesting that TNF-α participates in the mechanism through which c-Kit expression in the ICCs is inhibited via the NF-κB/miR-221 pathway.

IL-10 is an anti-inflammatory factor. Feng [7] performed pathological scoring of intestinal inflammation in EdnrB^−/−^ mice, showing that HAEC occurred mainly in the dilated segment, and found that the level of IL-10 in the stenotic segment of the colon of HSCR mice was higher compared with that in the dilated segment, suggesting that the inconsistencies in the severity of inflammation in the stenotic and dilated segments of the colon are associated with differences in IL-10 content.

Different expression levels of the abovementioned cytokines may serve to regulate the inflammatory response in the intestine, which, in turn, may lead to an impaired epithelial barrier, altered mucosal response healing, and disrupted mucosal immune and repair mechanisms, subsequently leading to the development of HAEC.

### 6.5. Inflammasomes

Inflammasomes are multiprotein complexes that are important components of the natural immune system and are involved in host-defense responses against a variety of pathogens. The most common inflammasomes are NLRP1, NLRP3, NLRC4/NAIP, NLRP12, NLRP6, pyrin and AIM2 [82].

Nakamura [83] identified significantly downregulated gene expression levels of NLRP3, NLRP12, NLRC4, ASC, and pro-IL-1, and significantly reduced protein expression levels of NLRP3, NLRP12, NLRC4, and ASC, in the colonic epithelium of patients with HSCR. Similarly, Tomuschat [84] found that the expression level of NLRP6 was significantly reduced in both HSCR aganglionic and ganglionic colon segments, and the relative expression level of NLRP6 was further reduced in patients who developed HAEC. Taken together, the above studies suggest that the downregulation of colonic inflammasome expression in patients with HSCR may lead to an altered colonic microbiome, which increases the patients’ susceptibility to developing HAEC.

Inflammasomes are able to mediate the classical pathway of cell pyroptosis. Li [85] explored the role of the LPS/miR-132/-212-sirtuin 1 (Sirt1)-NLRP3 regulatory network in HAEC. The results showed that LPS induced the upregulation of miR-132/-212, activated NLRP3 inflammasomes through suppressing Sirt1 expression, and promoted secondary cell pyroptosis in postoperative HAEC patients, in an LPS-stimulated HT29 cell line, and in LPS-treated mice. In addition, transfection experiments using an miR-132/-212 inhibitor and a Sirt1-overexpression vector resulted in a reduced rate of LPS-induced cell pyroptosis. However, the above effects were not sufficient to entirely eliminate cell pyroptosis, which supports the idea that other types of cell damage mechanisms may also operate. This suggests that LPS/miR-132/-212/Sirt1/NLRP3-Caspase-1 inflammasomes have a participatory role in the mechanism of HAEC progression: LPS induces miR-132/-212 upregulation, activates NLRP3 inflammasomes via suppressing Sirt1 expression and promotes cell pyroptosis, thereby promoting HAEC development and progression.

However, there are noted inconsistencies in the above findings, and the molecular mechanisms through which inflammasomes exert their functions have yet to be investigated in depth.

### 6.6. Exosomes

Exosomes are small membrane vesicles containing complex RNAs and proteins that are naturally present in body fluids, being involved in intercellular communication, body immune response, and antigen presentation [86,87,88]. Chen [89] isolated exosomes from the serum and found that the expression of miR-18a-5p was significantly increased in the HAEC exosomes, and that exosomal miR-18a-5p was responsible for the downregulation of RAR-related orphan receptor A (RORA), which activated the Sirt1/NF-κB signaling pathway and induced both apoptosis and the inflammatory response in intestinal cells, thereby promoting the development of HAEC.

In summary, the above studies collectively suggest the involvement of immune organs, immune cells, immunoglobulins, cytokines, inflammasomes, and exosomes in the development of HAEC (Figure 4), which, considered altogether, suggests that a close association exists between the immune system and HAEC.

## 7. Conclusions

There are several hypotheses that have been proposed concerning the pathogenesis of HAEC and the mechanisms that interact with each other (Figure 5), so further research should be comprehensive. Targeted therapies for HAEC could be developed in the future, starting from early regulation of intestinal microbiota, restoration of the intestinal peristalsis and mucosal barrier, and promotion of the intestinal immune system. At present, the treatment of HAEC is still mainly coordinated on symptomatic and supportive bases, and further research is required to properly elucidate the precise mechanism of its pathogenesis in order to assist with the treatment and prevention of HAEC.

## Figures and Tables

**Figure 1 ijms-24-04602-f001:**
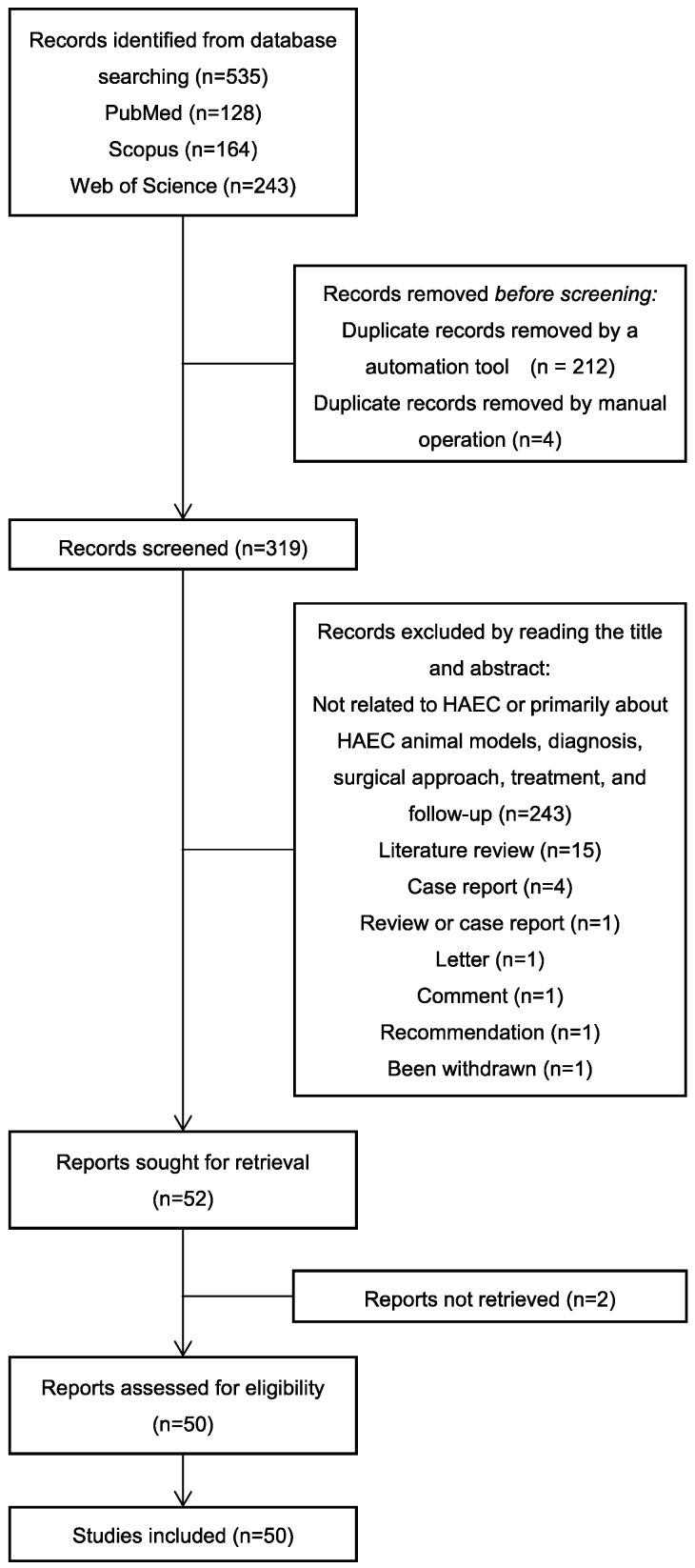
Flow chart: literature review.

**Figure 2 ijms-24-04602-f002:**
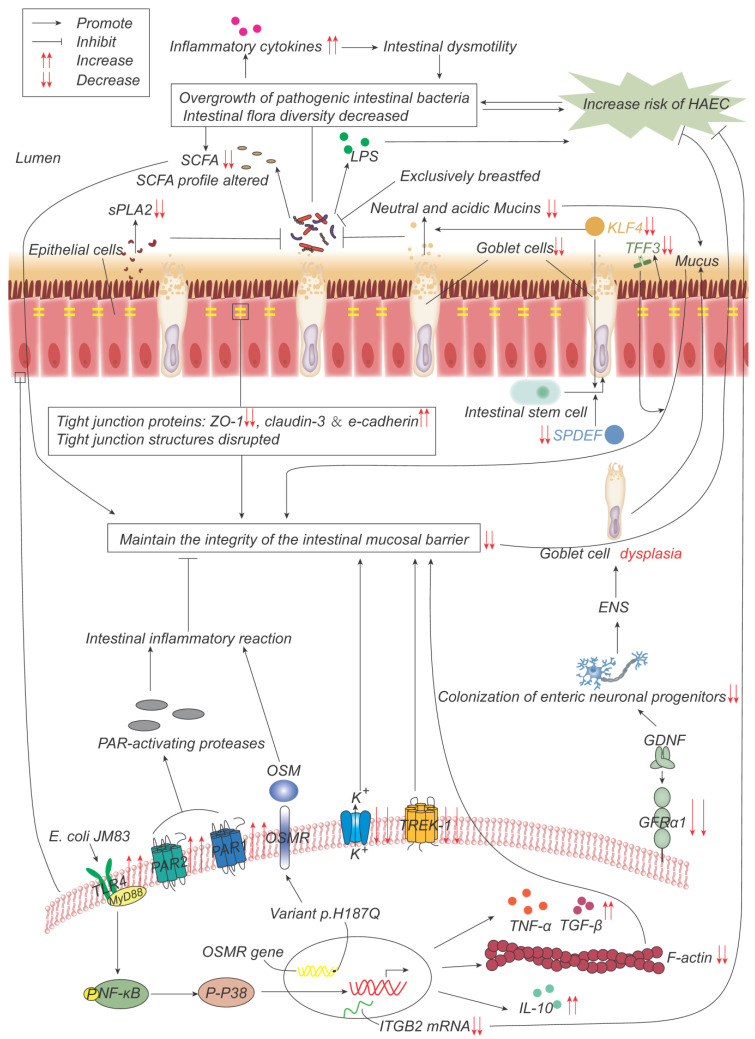
Pathogenesis advances in genes, intestinal microflora, and mucosal barrier. p.H187Q in the Oncostatin-M receptor (OSMR) gene is a susceptible variant of HAEC, which promotes inflammation by activating the OSM-OSMR *axis*. Decreased integrin beta-2 (ITGB2) mRNA expression is negatively correlated with the incidence and severity of HAEC. Overgrowth and reduced diversity of intestinal bacteria lead to increased release of inflammatory factors, which leads to intestinal dysmotility, which in turn leads to further bacterial overgrowth. Epithelial cells produce less secretory phospholipase A2 (sPLA2), which reduces the inhibition of bacteria and promotes bacterial overgrowth. Lipopolysaccharide (LPS) produced by bacteria can promote the development of HAEC, while exclusive breastfeeding regulates the gut microbiome in such a way that LPS production is reduced. Short-chain fatty acids (SCFAs) produced by bacteria are reduced, and their composition is altered, impairing the function of maintaining mucosal integrity. The expressions of TFF3, SPDEF, and KLF4 are significantly down-regulated, leading to the decrease in goblet cells (GCs) and the secretion of neutral and acidic mucins, which lead to the weakening of the mucosal barrier function. The expression of glial cell line-derived neurotrophic factor (GDNF) co-receptor and GDNF family receptor alpha-1 (GFRα1) is decreased, and the colonization of neuronal progenitor cells in the intestine is impaired, affecting ENS development, resulting in GCs dysplasia, and abnormal mucin production and storage. The structure of the tight-junction protein is damaged, and its composition is changed, which impairs its function of maintaining the mucosal barrier integrity. *Escherichia coli* JM83 stimulates NF-κB through TLR4 and MyD88, and through NF-κB/p-p38 signal transduction, F-actin protein density is significantly reduced, IL-10, TNF-α, TGF-β increase, leading to intestinal mucosal damage and promoting the development of HAEC. TREK-1 and K_(ATP)_ channels are reduced, leading to barrier dysfunction. Increased expressions of PAR-1 and PAR-2 lead to excessive local release of PAR-activating protease, which leads to inflammatory responses and impairs barrier function.

**Figure 3 ijms-24-04602-f003:**
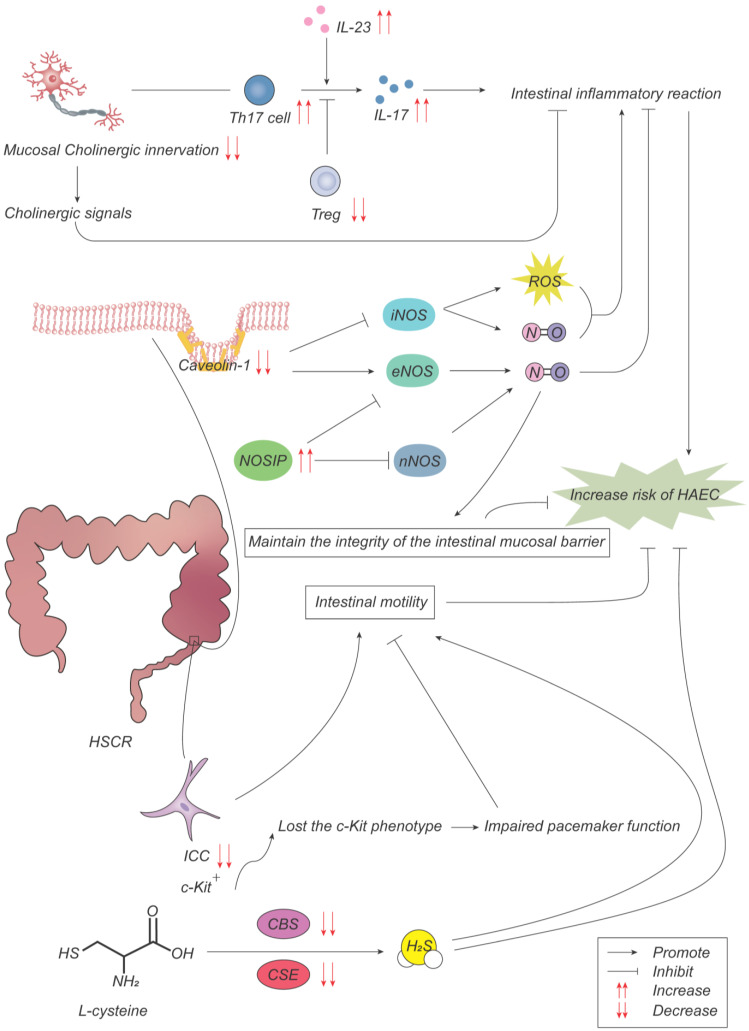
Pathogenesis advances in ENS. With a low cholinergic innervation degree in colon mucosa, Th17 cells and Treg cells regulate intestinal inflammatory response through the production of IL-17 and IL-23, promoting the development of HAEC. The decreased expression of caveolin-1 (Cav-1) results in the overactivation of inducible NO synthase (iNOS), which produces NO to promote inflammatory response while decreasing the inhibition of endothelial NOS (eNOS) on inflammatory response. The increased expression of NOS interacting protein (NOSIP) promotes inflammatory response and destroys the mucosal barrier by inhibiting NO production of neuronal NOS (nNOS) and eNOS. Interstitial cells of Cajal (ICCs) lose the c-Kit phenotype, resulting in impaired pacemaker function and intestinal motility. Cystathionine-β-synthase (CBS) and cystathionine-γ-lyase (CSE) are two key enzymes in the synthesis of hydrogen sulfide from L-cysteine. The decreased expression of CBS and CSE reduces gastrointestinal peristalsis and promotes the inflammatory response.

**Figure 4 ijms-24-04602-f004:**
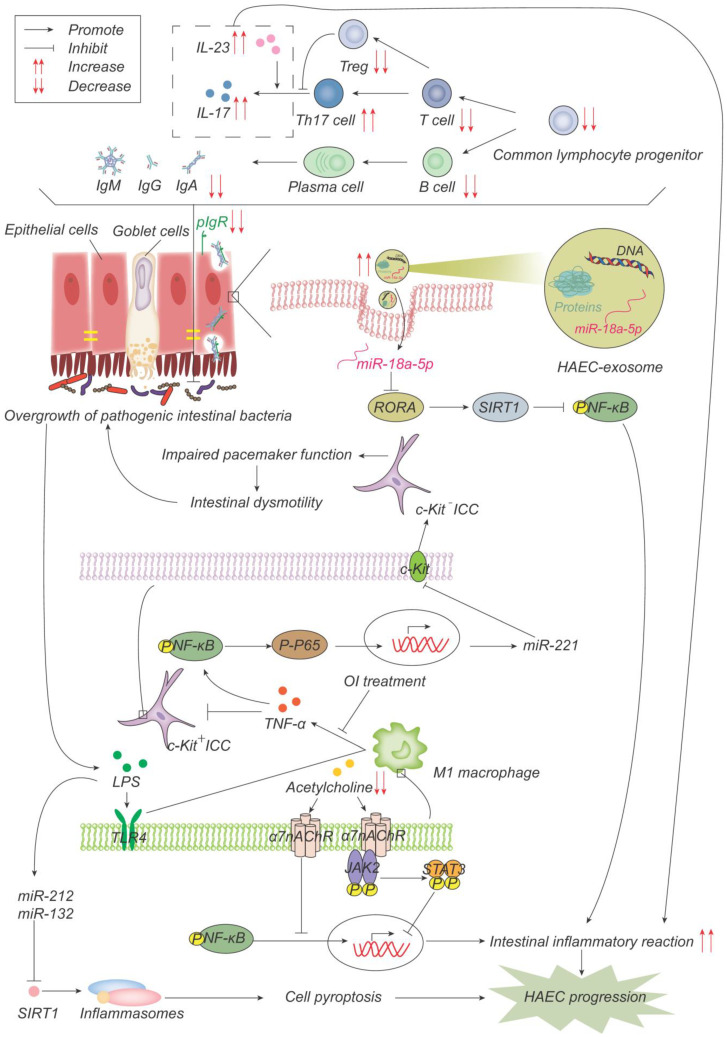
Pathogenesis advances in immunity. The number of common lymphocyte progenitor populations and, therefore, T cells and B cells decrease significantly, resulting in reduced immunoglobulin, which is actively transported to the mucosal surface by polymerized immunoglobulin receptors (pIgR) in the form of dimers to maintain the intestinal microenvironment balance and normal function. pIgR is also reduced, which together leads to intestinal microenvironment disturbances and bacterial overgrowth. Increased Th17 cells produce more IL-17 and promote the intestinal inflammatory response. On one hand, LPS produced by bacteria activates M1 macrophages through the TLR4 pathway, and M1 macrophages produce TNF-α, which inhibits c-Kit expression in ICC through the NF-κB/miR-221 pathway, leading to intestinal motility disorders, which further leads to the accumulation and overgrowth of intestinal bacteria. 4-octyl itaconate (OI) reduces the production of proinflammatory factors and promotes ICC phenotypic recovery by inhibiting macrophage activation. LPS, on the other hand, induces the up-regulation of miR-132/-212, activates inflammasome NOD-, LRR-, and pyrin domain-containing protein 3 (NLRP3) by inhibiting the expression of sirtuin 1 (Sirt1), promotes cell pyroptosis, and then promotes the occurrence and development of HAEC. Acetylcholine acts on α7 nicotinic acetylcholine receptor (α7nAChR) on the surface of macrophages, inhibits the activation of macrophages, activates the anti-inflammatory pathway of JAK2-STAT3 and suppresses the inflammatory pathway of NF-κB. Acetylcholine in HAEC is reduced, and the above anti-inflammatory effects are weakened. Exosome miR-18a-5p down-regulates RAR-related orphan receptor A (RORA), activates the SIRT1/NF-κB signaling pathway, induces excessive inflammatory response, and thus promotes the development of HAEC.

**Figure 5 ijms-24-04602-f005:**
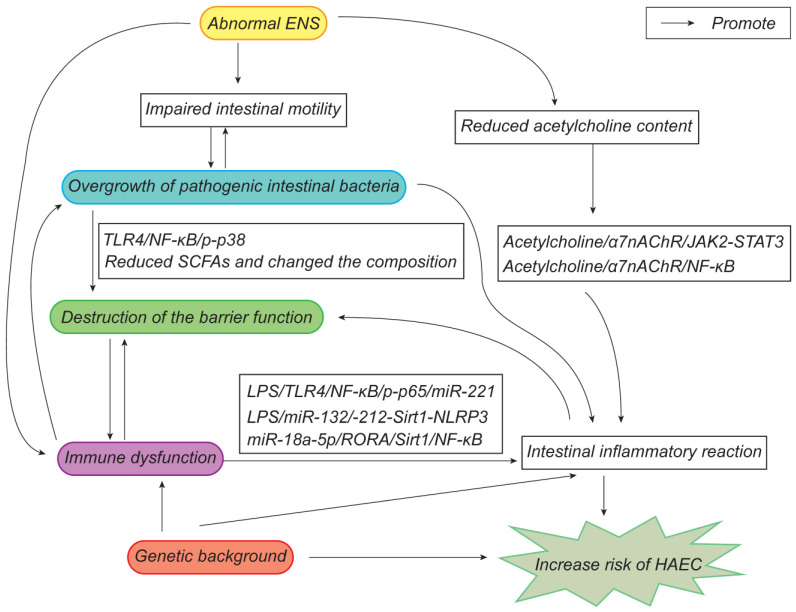
Updated pathogenesis of HAEC.

## Data Availability

The data presented in this study are available on request from the corresponding author.

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
