# Peer review of "Update on the Pathogenesis of the Hirschsprung-Associated Enterocolitis"

_ijms, 2023, doi:10.3390/ijms24054602_

Round 1

Reviewer 1 Report

Li et al. presented a review on the recent updates on the pathogenesis and  proposed molecular mechanisms of Hirschsprung disease-associated enterocolitis (HAEC). Overall, the review is well-written and covers multiple perspectives of pathogenic pathways, from genetic predisposition, microbiome, mucosal barrier dysfunction, intestinal dysmotility to abnormal inflammatory response. While a lot of the findings are related to the proximal/distal portion and aganglionic/ganglionic segments of the gut, the review can be further benefited by the inclusion of a short summary of the spatiotemporal hotspots of HAEC (e.g. position/time after surgery) in the introduction paragraph. In addition, the conclusion can end with an overall hypothesis and perhaps a figure to summarize how these different mechanisms may interact with each other in disease pathogenesis. Perspectives on the future direction to tackle HAEC both clinically and experimentally will be of interest to broad audiences.

Author Response

We really appreciate the kind comments and critical suggestions. We have included a summary of the spatiotemporal hotspots of HAEC in the introduction section and have added an overall statement of these hypotheses as well as an additional figure for illustration.

Reviewer 2 Report

Manuscript entitled „ Update on the pathogenesis of the Hirschsprung associated enterocolitis” is very interesting, well-written and well-planned review article. I fully support the publication of this manuscript; however I recommend the minor revision of manuscript. Small corrections should be made to the text according to the following comments:

Introduction

Line 51 - put a space after advances

1. Genes

Line 84 - put a space after background

Figure 2 - make a more detailed description of figure 2; explain with the full name of all the abbreviations that have been used in figure 2

Line 177 - put a space after microflora

Line 224 – should be write 70-80%

Line 249 – explain in full name an abbreviations K(ATP)

Line 283 – should be acetylcholine positive innervation

Line 369 - put a space after loss

Figure 3 - make a more detailed description of figure 3; explain with the full name of all the abbreviations that have been used in figure 3

Line 440 – explain in full name an abbreviation ELISA

Figure 4 - make a more detailed description of figure 4; explain with the full name of all the abbreviations that for the first time have been used in figure 4

References

Check the references and correct them according to the style preferred by the journal

Journal Articles:

1. Author 1, A.B.; Author 2, C.D. Title of the article. Abbreviated Journal Name Year, Volume, page range.

Author Response

Comment1:

Introduction

Line 51 - put a space after advances

  1. Genes

Line 84 - put a space after background

Line 177 - put a space after microflora

Line 224 – should be write 70-80%

Line 249 – explain in full name an abbreviations K(ATP)

Line 283 – should be acetylcholine positive innervation

Line 440 – explain in full name an abbreviation ELISA

Line 369 - put a space after loss

Answer: Thanks for the reviewer’s carefulness and patience. These mistakes were all corrected in the revised manuscript.

Comment 2:

Figure 2 - make a more detailed description of figure 2; explain with the full name of all the abbreviations that have been used in figure 2

Figure 3 - make a more detailed description of figure 3; explain with the full name of all the abbreviations that have been used in figure 3

Figure 4 - make a more detailed description of figure 4; explain with the full name of all the abbreviations that for the first time have been used in figure 4

Answer: Thanks for the reviewer’s constructive advice. We have added the description for figure 2, figure 3 and figure 4.

Comment 3:

References

Check the references and correct them according to the style preferred by the journal

Journal Articles:

  1. Author 1, A.B.; Author 2, C.D. Title of the article. Abbreviated Journal Name YearVolume, page range.

Answer: The references’ style has been checked and corrected.